# An Integrated Linkage Map of Three Recombinant Inbred Populations of Pea (*Pisum sativum* L.)

**DOI:** 10.3390/genes13020196

**Published:** 2022-01-22

**Authors:** Chie Sawada, Carol Moreau, Gabriel H. J. Robinson, Burkhard Steuernagel, Luzie U. Wingen, Jitender Cheema, Ellen Sizer-Coverdale, David Lloyd, Claire Domoney, Noel Ellis

**Affiliations:** 1John Innes Centre, Norwich Research Park, Colney Lane, Norwich NR4 7UH, UK; chie.sawada@jic.ac.uk (C.S.); carolj.moreau@gmail.com (C.M.); gabriel.robinson@infarm.com (G.H.J.R.); Burkhard.Steuernagel@jic.ac.uk (B.S.); Luzie.Wingen@jic.ac.uk (L.U.W.); jitender.cheema@jic.ac.uk (J.C.); claire.domoney@jic.ac.uk (C.D.); 2Institute of Biological, Environmental and Rural Sciences, Aberystwyth University, Plas Gogerddan, Aberystwyth SY23 3EB, UK; eys@aber.ac.uk; 3Germinal Horizon, Institute of Biological, Environmental and Rural Sciences, Aberystwyth University, Plas Gogerddan, Aberystwyth SY23 3EB, UK; david.lloyd@germinal.com

**Keywords:** pea, genetic map, recombinant inbred population, integrated map

## Abstract

Biparental recombinant inbred line (RIL) populations are sets of genetically stable lines and have a simple population structure that facilitates the dissection of the genetics of interesting traits. On the other hand, populations derived from multiparent intercrosses combine both greater diversity and higher numbers of recombination events than RILs. Here, we describe a simple population structure: a three-way recombinant inbred population combination. This structure was easy to produce and was a compromise between biparental and multiparent populations. We show that this structure had advantages when analyzing cultivar crosses, and could achieve a mapping resolution of a few genes.

## 1. Introduction

The three recombinant inbred line (RIL) populations derived from reciprocal crosses between the cultivars Brutus (B), Enigma (E), and Kahuna (K) were generated to provide a set of populations derived from intercultivar crosses of interest to stakeholders of the UK Pulse Crop Genetic Improvement Network [1]. Analysis of early generations of the populations identified genetic regions associated with some traits of interest to pea breeding programs (yield, standing ability, seed size) [1]. Here, we describe a further analysis of these populations, using advanced RILs that were genotyped using 13,204 biallelic SNPs anchored to the pea genome sequence [2], and show the benefit of treating this as a single population for the purposes of genetic mapping.

## 2. Materials and Methods

### 2.1. Plant Material

The parents of the segregating populations were the cultivars Brutus (B), Enigma (E), and Kahuna (K). The source of plants used and their relationship to other cultivars are described in Moreau et al. [1]. Brutus AFP 84/384 was bred by DLF Seeds Ltd., and its parents were DP2459/86 and Bohatyr. Enigma was bred by Toft Plant Breeding, and its parents were the cultivars Jackpot and Delta. Jackpot was derived from a cross between Bohatyr and Solara. Brutus and Enigma therefore have some common genetic background derived from the cultivar Bohatyr. Kahuna was also bred by Toft, but with Cebeco as the UK agent, and this variety is currently listed in the UK under Limagrain (pers. comm. with Jens Christian Nørgaard Knudsen of Nordic Seed and Steve Belcher of PGRO). The RILs derived from the three-way crosses described [1] were advanced by single-seed descent to F_13_, after which seeds were bulked to support replicated glasshouse and field studies. The three sets of advanced RILs were sown under glasshouse conditions using three or four replicates per line, and phenotypic data were gathered. Seeds were harvested from completely dried and senesced plants.

### 2.2. SNP Data and Mapping

RILs and their parents were genotyped by Neogen UK, using an Infinium array that detected 13,204 biallelic SNPs. These SNPs were either monomorphic or differed such that two cultivars shared an allele and the third was different. Thus, it was expected that any segregating marker would be scored in two of the three populations, creating the possibility of integrating the maps of these three RIL populations directly, and allowing a more precise association of allelic and phenotypic differences.

Accordingly, the data from the three RIL populations were collated in an Excel file so that all of the data for a single marker were represented as a single row. Each column corresponded to a RIL, and the score was recorded as “B”, “E”, “K”, or “-”, corresponding to the parental origin of the allele, and “-” representing missing data.

Note that for a marker, these three codes refer to just two alleles, but this scoring scheme records the parent of origin of the allele, which is required for genetic mapping. For example, the marker PsCam004787_3611_167 (the top marker on linkage group I) detected a T/C SNP and segregated in BK and EK but not BE, so the B allele in this case was identical to the E allele and different from K. The three possibilities are shown in Table 1.

### 2.3. Estimates of Relatedness

The fraction of alleles shared between each pair of the parents was calculated directly from the data presented in Table 2. If dBE, dBK, and dEK represent the fractional distances between Brutus and Enigma, Brutus and Kahuna, and Enigma and Kahuna, respectively, then, in Figure 1, the length of the line from the point of trifurcation to Brutus (dBt) is ½(dBE + dBK – dEK), etc. Statistical tests of deviation between observed and expected number of marker differences for each linkage group were performed as χ^2^ tests in which the expected value was in proportion to the total value for all linkage groups.

### 2.4. Construction of Individual Genetic Maps Using ASMap

For genotype data preparation, redundant markers were removed, and initial clustering of markers according to the pairwise Hamming distance between markers was carried out by ThreadMapperStudio [3] for each population. Each cluster was allocated a linkage group (LG) number in line with Table S3 of [4], and markers were labeled with this LG number in the genotype file. The library “ASMap” (v1.0-4) [5] from the R software environment (v4.0.3) was employed for the genetic map construction using a custom-made script (available at https://github.com/wingenl/genetic_mapping_with_ASMap accessed 24 December 2021). Default parameters were applied for the mapping, with the exception of the *p*-value threshold of 10^−9^ for the clustering and map distance calculation using the Kosambi mapping function. Subsequently, the orientation of each linkage group was aligned in the direction indicated in Table S8 of [4].

### 2.5. Construction of An Integrated Map

A total of 4556 markers were mapped in these populations, as summarized in Table 2. A small number of markers were mapped in only one population due to technical problems with allele calling.

Neogen allele calls were converted to “B”, “E”, and “K”, scores as shown above. The positions of 4556 PsCam markers on the seven pseudomolecules of the Caméor v1a assembly were obtained with a blastn analysis using marker sequence information from Supplementary Table 2 of [4] and the Caméor v1a Jbrowse site (https://urgi.versailles.inra.fr/Species/Pisum accessed 24 December 2021). Markers positioned on a pseudomolecule were ordered according to their physical position, and this was used as an initial order for the genetic map. This order was then examined to see whether local rearrangement might minimize the number of close double recombinants that were proposed by the initial order. Markers assigned to scaffolds, or that were not in a position consistent with the genetic map, were positioned with respect to their closest marker. Essentially, this was the colormapping method of Kiss et al. [6]. The final map data are available in Appendix A, and a heat map of all possible pairwise distances between markers within linkage groups is shown in Appendix A.

### 2.6. Single-Marker Analysis of Quantitative Traits

The average weight of the seed from each RIL was obtained from greenhouse grown plants. These values were then normalized so that each of the three RIL populations had a mean and standard deviation of 0 and 1, respectively. For each marker, the mean and standard deviation of the normalized values was calculated, and a *t*-test was performed to estimate the significance of the deviation of this difference from the expected value of 0.

## 3. Results

### 3.1. Relatedness of the Parental Genomes

Considering only the 4497 markers that were scored in two populations (i.e., where the allelic state was known in the three cultivars), the degree of relatedness between the parents could be estimated, and how this differed among the linkage groups is illustrated in Figure 1. Overall, Kahuna was more distinct from Brutus or Enigma than these were from each other. There were about twice as many markers that distinguished Kahuna from Brutus and Enigma than distinguished either of these from the other two (Table 2). However, this differentiation was not uniformly distributed among linkage groups (Table 2, Figure 1). Kahuna was most distinct for linkage groups IV and VII; while for LG I, it was most similar to the other two lines. Enigma was notably distinct for LG V, and this was statistically the most significant difference (as compared to the overall values).

### 3.2. The Number of Recombination Events Per RIL

In pea, there are ca. 16 chiasmata per meiosis [7], as crossing over occurs at the four-strand stage. This means that each gamete has, on average, ca. eight crossover events, so in total ca. 16 per individual sporophyte. In the construction of RILs, there were multiple meioses, so in each gamete in each generation, there were ca. eight crossovers. In RILs derived from selfing, in which the fraction of gametes that exhibit a recombination event is *R*, the corresponding fraction of recombination events per meiosis is *r*, and is given by the relationship *r* = *R*/2(1 − *R*) [8]. The frequency distribution of recombination events in the RILs was therefore related to the number of crossovers per meiosis and the number of meioses. If a crossover occurred in a region that was homozygous (distally from the centromere towards the telomere), then the consequences of this event were not observable. Only ca. ½ of crossovers in the F_2_ led to observable recombination, ¼ in the F_3_, and so on. The total number of recombination events seen to accumulate in the gametes of a RIL, after endless generations, was about twice the number of such events from the F_1_ meiosis. After endless inbreeding, all gametes from a RIL were identical, so the number of observable recombination events in an RIL was about the same as in an F_2_; however, in the RIL, they were in the same location in the maternally and paternally derived genomes. Thus, in pea, we expected to see about 16 crossover events per RIL, and the frequency distribution of these would approximate to a Poisson distribution (for which the mean was equal to the variance), as the recombination events were independent of one another. The distribution might have been slightly tighter than a Poisson, because at least one chiasma was required per bivalent for proper chromosomal disjunction (i.e., two out of the four gametes had a recombination event). The observed frequency distributions of recombination events per RIL are shown in Figure 2, and the means and variances are given in Table 3.

### 3.3. The Linkage Map

The length of each linkage group was determined by counting the number of recombination events between adjacent markers, and then using these data to determine the fraction of RILs (*R*) that had recombined the parental alleles for the adjacent markers; from this value *r*, the recombination rate per meiosis was determined as above. The recombination rate was converted to map distance using Haldane’s mapping function [9], which assumed that recombination events were independent of one another. The correlation coefficient between map length (cM) and the average number of recombination events (*R*) per linkage group was moderate (*r*^2^ = 0.55). Regression analysis gave the relationship cM = 42.4 + (49.3)*R*, consistent with a requirement for one recombination event per chromosome (enabling proper disjunction) and for each recombination event corresponding to 50 cM. The compiled linkage map is shown in Figure 3.

Extended regions (>10cM) that did not have segregating markers in any of the three populations are marked with a dashed line in Figure 3. These were likely to be regions that were identical by descent (IBD), and may include genes that were selected by breeders; potential candidate genes for these selected traits are discussed below.

**Linkage group I**. All three parents were *afila* mutants, i.e., they were homozygous for a recessive *af* allele, and this was consistent with the large gap in the map in the region of *Af* on LG I; this further suggested that these three varieties carry the same *afila* allele.

**Linkage group II.** All three parents were white-flowered and carried a recessive allele at the *A* locus, yet there was no signature of IBD in this region. The data suggested that the *a* allele of Brutus had a different haplotype from Enigma or Kahuna. There were two large gaps (of presumed IBD) in this linkage group, but only the region including the genes *Late1*, *Rug3,* and *K* had identified genes, and of these, *Late1* seemed the most likely to be under selection, as it is involved in photoperiod response [10]. Interestingly, for the *Lf* gene [11], which is also involved in the regulation of flowering time, but in a different pathway [12], Brutus and Kahuna shared a haplotype that was distinct in Enigma.

**Linkage group III**. This linkage group had two extended IBD regions. One was centered around *Hr*, another gene involved in the short-day response of pea [13], again suggestive of selection in relation to the control of flowering time. Interestingly, the genes *Dne* (LG III) and *Sn* (LG VII), also involved in this pathway [13], did not appear to share identity by descent.

**Linkage group IV.** Although there were several large IBD gaps in this linkage group, there was no obvious selection target identified.

**Linkage group V**. The single large IBD region in this linkage group was associated with the genes *Det*, *R,* and *Tl*, but these were not centrally located in the gap. All three cultivars were round seeded (and so carried a dominant allele at *R*), but it would seem unlikely that the same *R* allele would be under selection in independent breeding programs. Berdnikov [14] identified a lethal mutation in the *R–Tl* interval; this unidentified gene may be under strong selection, but it seems most likely that the locus under selection by breeders has not been identified.

**Linkage group VI**. There were two putative IBD regions in this linkage group. The largest of these, at the top of the linkage group, did not appear to be associated with a clear candidate for selection, but the lower segment was associated with the gene *ANR1a*, which is involved in root development [10], and is a possible candidate.

**Linkage group VII**. This linkage group was notable for the lack of markers segregating in the BE cross; those few markers showing allelic differences between Brutus and Enigma were in three tightly clustered groups (Figure 1 and Figure 3, Table 2). Statistically, this was not the most significant deviation from the expected number of markers, but the combination of their number and marker distribution suggested that most of this linkage group was shared by Brutus and Enigma. This meant that the presumed IBD region for linkage group VII reflected the similarity of two rather than three chromosomes. The same could be argued for linkage group IV, which had a greater statistical significance, but the markers distinguishing Brutus and Enigma were more distributed. This similarity was probably due to shared descent derived from the cultivar Bohatyr (see the Materials and Methods section).

The genes identified as being associated with putative identity by descent regions among these three cultivars were all associated with phenology, flowering time, and plant architecture. This may simply reflect the nature of those genes that have been most intensively studied and were assigned positions on the genetic map. However, it is interesting that those genes assigned positions on the map that were associated with disease resistance (*Dmr*, *Er1*, *Sbm1,* and *Ppi2*) were not associated with IBD regions.

Of the pea NBS LRR genes discussed in [15], only one, AF123702, fell within an IBD region, on linkage group VI close to *Lst*. There were 225 positions on the “Caméor” v1a assembly that corresponded to NBS LRR gene sequences, and an additional 22 mapped to scaffolds (Appendix A). Of the 225 that mapped to the pseudomolecules, 12 mapped to positions corresponding to IBD regions. The total map length was 929 cM, and the 13 IBD regions had a total map length of 250 cM, so by chance alone, we would expect 61 NBS LRR genes to lie within these gaps. The underrepresentation of NBS LRR genes within these gaps was statistically significant (χ^2^ ≃ 52), and may have reflected breeding selection for diversity at these sites.

### 3.4. Recombination: Map Distance between Recombination Events

A statistical treatment of the frequency distribution of recombination events along a linkage group based on the distribution theory of runs [16] is given in Appendix B. The data, which are plotted in Figure 4, showed a good fit to the expected distribution, which was consistent with recombination events being distributed at random with respect to the genetic map.

### 3.5. Comparison to the Three Individual Genetic Maps

Marker order and linkage group length were broadly conserved between the integrated map and the three separate maps (Appendix A). There were two main differences, the first was that the estimation of length of the putative IBD regions was more difficult where the gaps were longer, and the order of the segments was not easily determined when the gaps were of the order of 50 cM. In addition, there were regions where the marker order was inverted with respect to the integrated map and other individual maps. These were always adjacent to a large gap in the individual map, and presumably the mapping program minimized the number of recombination events between segments of the map that were very distant, and consequently added some double recombinants at the end of a linkage segment (Appendix A). Note that these were inversions of marker order in the map, not chromosomal inversions.

### 3.6. Genetic and Physical Maps

In general, the relationship between the genetic map and pseudomolecule assembly followed a sigmoid curve (Figure 5). The nearly vertical sections, where the recombination rate per Mb was very low, corresponded to pericentric regions [2]. Outside these regions, there were extended regions with a slope of roughly 1–3 Mb/cM. The gaps in the curves corresponded to regions of the pseudomolecule where no markers segregated in any of the three RIL populations; these were regions presumed to be identical by descent. These gaps did not seem to correspond to regions with exceptional recombination rates in comparison with their surrounding regions. This supported a recent coancestry of these regions, rather than exceptionally low rates of recombination.

Linkage group V, chromosome 3, had the lowest standard deviation of the cM/Mb ratio, and the recombination rate corresponded to approximately two crossovers per chromosome, but these were unevenly distributed among the two chromosome arms. For the long and short arms, the cM/Mb ratios were 0.35 ± 0.25 and 0.66 ± 0.39, respectively. The long arm was ca. 100 cM, and the short arm ca. 25 cM in length; the long arm had almost four times the number of crossover events of the short arm, and when taking this into account, the recombination rate per unit length of the long arm was about half that of the short arm. This chromosome therefore somehow distinguished the recombination rate on either side of the centromere. This chromosome was reported to have unusual synapsis in the cross JI15xJI399 [7], with severely reduced crossover number, which could be related to the difference in observed recombination rate between both chromosome arms.

### 3.7. Nonlinear Patterns of Alignment

Linkage group II (chromosome 6) and linkage group VII (chromosome 7) both had extended regions where the mapping between linkage group and pseudomolecule was disturbed (Figure 5). In these regions, markers that were genetically close to each other were far apart on the corresponding pseudomolecule. In both these cases, nonlinearities corresponded approximately to the position of the centromeres. These could be explained as a consequence of misassembly of regions that had a high density of retrotransposons (notably, Tekay and Ogre [2]). Misassembly of the pseudomolecule was not the only possible explanation of this behavior, and it should be noted that these major disruptions to linearity all occurred close to the centromere, where non-recombination has been interpreted as a lack of crossover, but could in fact correspond to double recombination events within the centromere and occasionally in the adjacent segment. Such events would lead to a misassembly of the genetic map.

A second type of nonlinearity in the relationship was seen in the small number of scattered points where the order of markers in the genetic map was different from the order of the corresponding sequence in the pseudomolecule. These may have been errors in either the genetic map or the assembly, and in addition, could be a consequence of gene conversion events, which would appear as close double recombination events in the genetic map.

### 3.8. Quantitative Traits

The integrated map comprised 621 recombinant inbred lines, so the minimum fraction of lines that were recombinant (*R*) was ca. 1/600, corresponding to a recombination fraction per meiosis (*r*) of ca. 0.08 cM [8]. As the total map length was ca. 900 cM (Table 4), this meant that the resolution was ca. 0.01%, corresponding to a small number of genes. This suggested that the map had the potential to identify candidate genes for quantitative traits.

Previous work with these three populations [1], and a map mainly derived from retrotransposon-based, sequence-specific amplified polymorphism (SSAP) genetic markers, identified QTL for seed size (mean seed weight) on linkage groups I (two loci), III, IV, and V. Using mature seed weight data for the advanced RILs (Appendix A), a single-marker analysis of seed weight differences again identified QTL on these linkage groups.

The largest QTL on LG I (Figure 6) close to AGPase [1] (PsCam060155) was identified in the BK subpopulation. The corresponding QTL in the EK subpopulation was in a slightly different location, close to a gene annotated as a transmembrane amino acid transporter (Appendix A). Previously identified QTL on the other end of LG I and LG III in the BE subpopulation (Moreau et al. [1]) were not confirmed by these data. The QTL on LG IV was at PsCam026873, annotated as “Annexin family signature”, and the peak on LG V was at PsCam054756, annotated as “Eukaryotic translation initiation factor 3 subunit 7 (eIF-3)”. The lowest Student’s *t*-value for the BE subpopulation was at PsCam001302, but this was likely to be an artifact, as there were many missing scores for this marker, and it cosegregated with 17 other markers that spanned a 30 Mb region in the pericentric region of chromosome 1 (LG VI, Figure 5). Nevertheless, this block as a whole did have the lowest Student’s *t*-value for BE, suggesting that there was indeed a gene within this pericentric block for which allelic variation contributed to variation in mean seed weight.

## 4. Discussion

We have shown that a set of recombinant inbred line populations derived from all pairwise combinations of crosses between three inbred lines could provide useful resources for genetic mapping. This strategy allowed the genetic variation among the three inbreds to be studied coordinately, and the combination of three parents increased the number of markers that could be scored. In addition, the three RIL populations facilitated the generation of a large number of informative RILs, and therefore more overall recombination. Mapping with the integrated populations mitigated some difficulties with determining marker order in individual biparental RIL populations, as was exacerbated by regions of presumed identity by descent. The problem of identity by descent is inevitable in the examination of crosses between related cultivars. Indeed, the differences in marker order as determined by ASMap for the individual subpopulations and the integrated map were all associated with regions of pairwise identity (Appendix A). The three-way recombinant inbred population structure thus increased both the marker density and the resolution of the genetic map, while the inclusion of a third parent could compensate for some of the difficulties that could arise due to identity by descent.

In this population, we identified several regions of putative identity by descent among all three parents, and pairwise IBD differed among linkage groups (Figure 1 and Figure 3). We ruled out that these regions were caused by an unusual distribution of recombination events (Figure 2 and Figure 6) to support their identity by descent. We hypothesized that breeder selection led to regions identical by descent, and suggested plausible gene candidates as the targets of selection for some regions (Figure 3); however, many IBD regions currently lack gene candidates. The most obvious example of an IBD region explained by a candidate gene was the region encompassing the *Afila* gene; all three cultivars were *afila* mutant types, and presumably carried the same allele at this locus, accounting for the extended region of identity by descent. However, the behavior of the region around the gene *A* was puzzling. All three cultivars were homozygous for *a* and were white-flowered; however, the data suggested that recombination occurred close to the *A* locus in the ancestors of these lines, generating new haplotypes in this region. The contrast between *A* and *Af* probably reflected the antiquity of the introduction of white-flowered cultivars [17].

A close examination of the genotypic data showed that recombination events were randomly distributed among RILs. Within the RILs, these were also randomly distributed with respect to the genetic map (Figure 2 and Figure 4). However, this was not the case with respect to the physical map, as we saw extended regions lacking recombination corresponding to the pericentromeres, as has been reported previously [2]. Outside these regions, the recombination rate per unit physical distance was reasonably constant (Figure 6 and Appendix A).

The high marker density and the number of RILs in the combined population enabled a simple single-marker-based approach to determine candidate genes underlying quantitative traits as illustrated by seed size (Figure 6 and Appendix A).

These observations suggested that three-way recombinant inbred populations are a powerful tool for the analysis of genetic variation, especially among cultivated material for which identity by descent can make genetic map construction problematic. For species such as pea, where seeds are large and seed multiplication rates and the number of seed set per cross pollination are low, the production of multiparent advanced generation intercross populations [18] is very labor-intensive, while the production of recombinant inbred lines is relatively easy once the initial crosses have been performed. The three-way recombinant inbred combination added more genetic diversity to a mapping population compared to a single RIL population, and the generation of ca. 600 RILs was readily achievable, providing mapping resolution of the order of a few genes.

## Figures and Tables

**Figure 1 genes-13-00196-f001:**
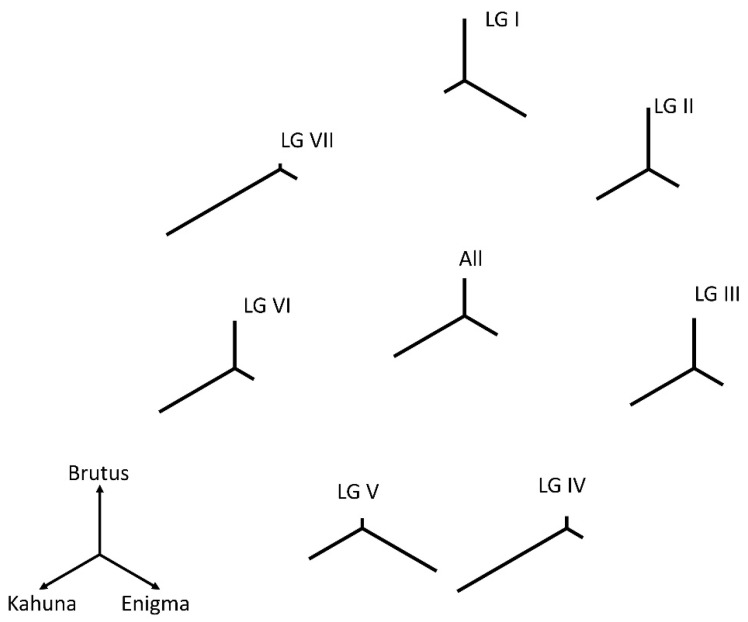
Genetic distances between parents. A representation of the genetic differences between Brutus, Enigma, and Kahuna. The relative genetic distances are represented by three linear vectors, each at an angle of 120° with respect to the other two. The differences between individual linkage groups (labeled LG I to LG VII) and the overall difference (labeled All) are given. The sum of the linear vector lengths was constant.

**Figure 2 genes-13-00196-f002:**
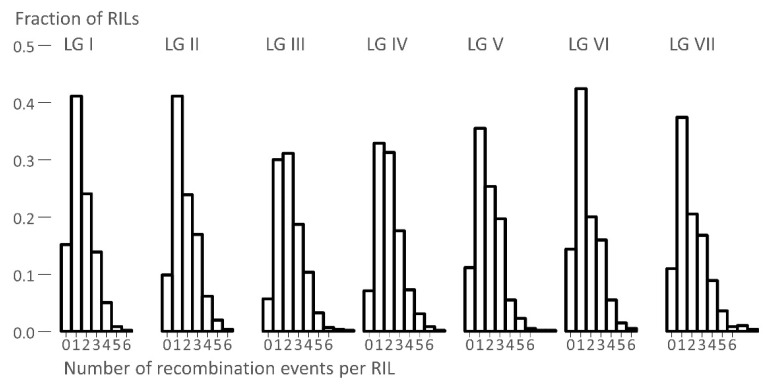
The frequency distribution of the number of recombination events per linkage group. The number of crossover events per RIL is plotted for each linkage group on the *x*-axis, and the frequency of that number of crossover events per RIL is plotted on the *y*-axis.

**Figure 3 genes-13-00196-f003:**
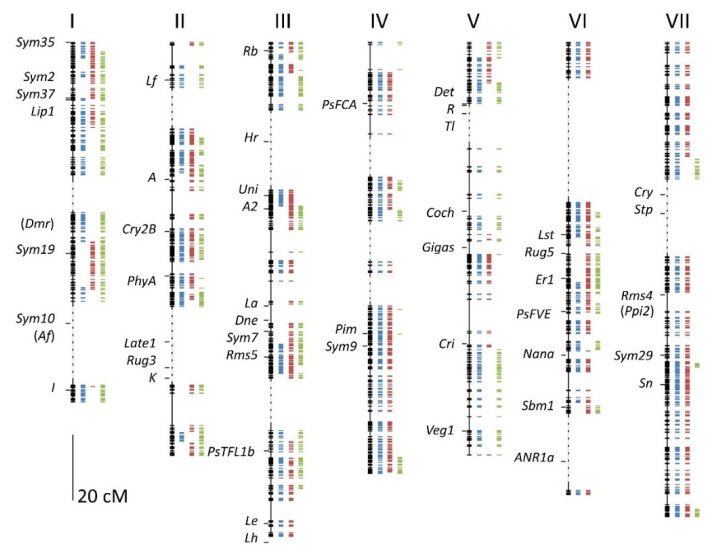
The integrated genetic map of three RIL populations. The position of genetic markers are indicated as short horizontal bars on the combined linkage map, with horizontal bars to the right of each linkage group indicating the population in which the markers segregated (blue, red, and green for the EK, BK, and BE populations, respectively). Extended segments of the map, of at least 10 cM devoid of markers, which were most likely identical by descent, are indicated by dashed lines; the lengths of these regions are determined by the recombination fraction between the flanking markers. Genes corresponding to classical morphological or physiological phenotypes identified on the pseudomolecules of the Caméor v1a sequence assembly [2] are indicated by name; see Appendix A. Approximate positions of genes placed on the map through linked markers are indicated by the gene symbols in brackets.

**Figure 4 genes-13-00196-f004:**
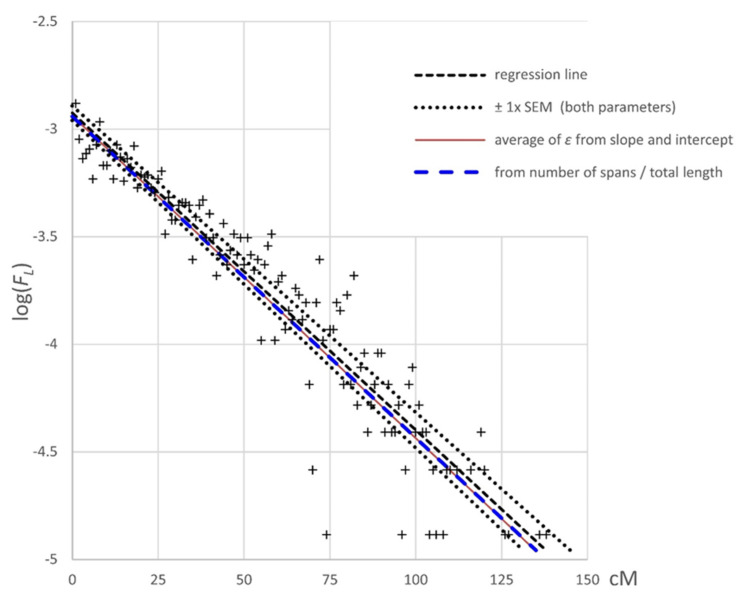
The frequency distribution of map distances between recombination events. The expectation for the frequency distribution of map distances between adjacent recombination events in a given individual is described in Appendix B. This distribution is defined by the relationship $\log F_{L}=2\log\left(\epsilon \right)+L\log\left(1-\epsilon \right)-\epsilon \log\left(1-\epsilon \right)$, where *F_L_* is the frequency of intervals of length *L* cM and ε (=*L*/*n*) (see Appendix B). The regression coefficient for the fitted line was 0.94, and the regression line ± one SEM is plotted on the graph. The expected relationship from the number of intervals observed and the total length of these intervals is plotted as the blue dashed line, and the solid red line represents the predicted relationship from the average value of ε from the slope and intercept of the regression line.

**Figure 5 genes-13-00196-f005:**
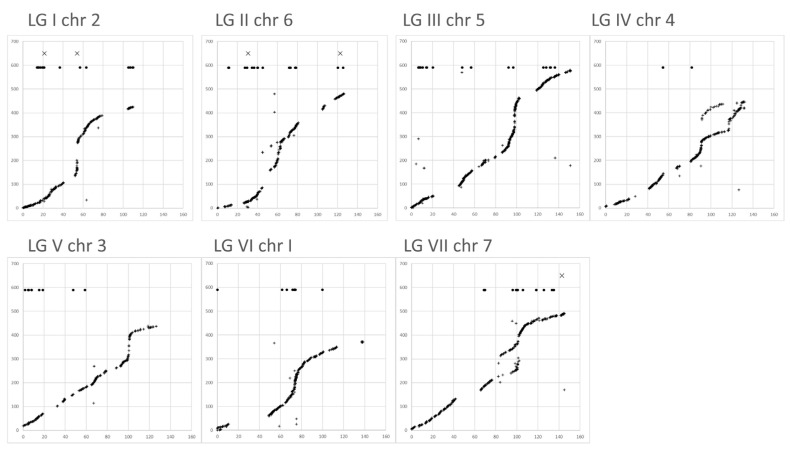
Comparing genetic maps with chromosome assemblies. For each linkage group, marker position in cM is plotted on the *x*-axis against the position of the corresponding sequence on the cognate pseudomolecule (in Mb on the *y*-axis). In total, 4556 markers were plotted. The five markers that mapped to positions on other pseudomolecules are marked “X” at the top of each graph, and 92 markers that mapped to scaffolds are marked as filled circles at the top of each graph.

**Figure 6 genes-13-00196-f006:**
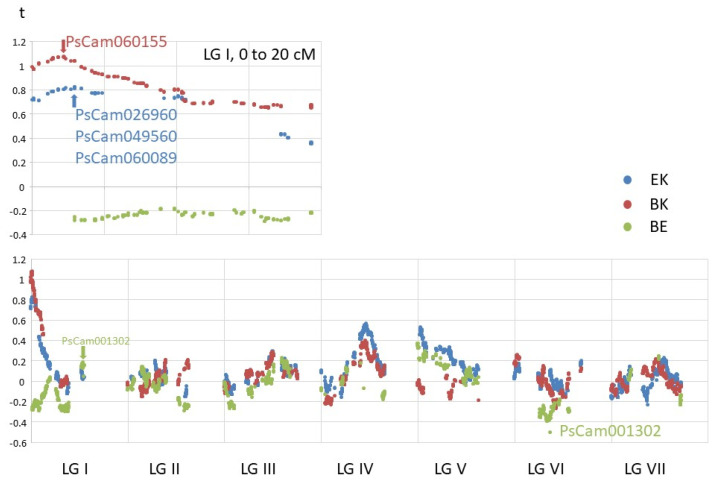
Mean seed weight analysis. Mean seed weight data were obtained for all RILs (except for one RIL, BK3). These were standardized to a mean of zero and standard deviation of 1 within each RIL population. For each marker, the difference between the means of RILs with contrasting genotypes was calculated, and then the Student’s *t* statistic was calculated, and this value was plotted on the y-axis, with marker position on the x-axis. Genes close to the maximum or minimum *t* for each of the three subpopulations are indicated (see also Figure 5). The upper plot shows an expanded linkage group I.

**Table 1 genes-13-00196-t001:** Examples of the three types of allelic calls.

Marker	SNP	Segregates in			Common	Identity
PsCam004787	T/C		BK	EK	K	B = E
PsCam014211	T/G	BE		EK	E	B = K
PsCam042135	T/G	BE	BK		B	E = K

**Table 2 genes-13-00196-t002:** Distribution of markers among populations.

	Number of Segregating Markers Scored											
	By Population					Unique to				Shared by		
	Total	EK	BK	EB		EK	BK	EB		EK & BK	EK & EB	BK & EB
LG I	550	327	295	461		7	7	3		75	245	213
LG II	726	438	560	444		2	7	1		273	163	280
LG III	975	664	768	518		0	0	0		457	207	311
LG IV	605	556	530	118		0	4	2		483	73	43
LG V	456	425	206	276		3	1	1		176	246	29
LG VI	576	398	491	254		2	7	0		313	83	171
LG VII	668	637	580	107		5	4	3		552	80	24
total	4556	3445	3430	2178		19	30	10		2329	1097	1071

The numbers of markers per linkage group (LG) are shown for the three populations. Numbers of markers that were unique or shared among populations are given. The shaded entries are those for which the number of distinguishing markers was close to the expectation from the data for all linkage groups; i.e., where the χ^2^ value was less than 10.

**Table 3 genes-13-00196-t003:** Mean and variance of the number of recombination events per RIL linkage group.

Linkage.		Number of Crossovers						
Group		Mean		Variance		Maximum		Variance/Mean
I		1.56		1.26		6		0.81
II		1.76		1.39		6		0.79
III		2.13		1.64		8		0.77
IV		1.98		1.52		7		0.77
V		1.83		1.57		8		0.85
VI		1.62		1.44		6		0.89
VII		1.96		2.14		8		1.09

**Table 4 genes-13-00196-t004:** Recombination rates per linkage group and chromosome.

	Total Length					Number of		cM/Mb		
LG	cM		Mb			Marker Intervals		Mean		SD
I	110		342	ch2		490		0.52		1.70
II	126		384	ch6		611		0.41		0.71
III	151		463	ch5		832		0.32		0.32
IV	132		357	ch4		544		0.52		2.07
V	126		438	ch3		442		0.36		0.29
VI	138		298	ch1		508		0.40		0.76
VII	145		393	ch7		583		0.46		0.68
All	928		2675			4010		0.42		1.09

Table 4. The length of the pseudomolecules of the Caméor v1a assembly, taken from https://urgi.versailles.inra.fr/Species/Pisum (accessed 24 December 2021) and not including scaffolds not assigned to pseudomolecules; the corresponding chromosomes are designated ch1 to 7. The recombination rate along each linkage group in cM/Mb was calculated as the mean of the difference in position in cM divided by the difference in position (in Mb) for markers separated by 10 marker intervals. For some pairs, this could not be calculated because the markers were too close to the end of a linkage group, at least one marker was on a scaffold or other pseudomolecule in the v1a assembly. For those markers on the same pseudomolecule that were anomalously far apart, and where this difference was negative, the marker pair was ignored; where the difference was positive, this contributed to an anomalously low cM/Mb value. The recombination rates are plotted in Appendix A.

## Data Availability

Date are presented in the excel file comprising of Appendix A.

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
