# Peer review of "An Integrated Linkage Map of Three Recombinant Inbred Populations of Pea (Pisum sativum L.)"

_genes, 2022, doi:10.3390/genes13020196_

Round 1
Reviewer 1 Report
In this study, authors described three sets of advanced RILs population comprising 621 recombinant inbred lines, derived from the three-way crosses from the pea cultivars Brutus, Enigma and Kahuna. The populations were genotyped by 13,204 SNPs anchored to the pea reference genome and a total of 4556 markers were mapped. An integrated genetic map with total map length of 928 cM was constructed. Using seed weight data for RILs, a single marker analysis identified QTL on linkage group I, III , IV and V. Several candidates for the QTLs controling pea seed size were identified, suggesting that the high marker density and the number of RILs in the combined population has the potential to clone candidate genes for important traits. These results would provide helpful information for molecular cloning and marker assisted selection in pea. I suggest the linkage group in the manuscript should be abbreviate to LG (not lg).
Author Response
We have changeg 'lg' to LG' throughout.
Reviewer 2 Report
Thank you for carrying out this truly excellent research.
I think a more extensive introduction would benefit the manuscript because of the broad readership. Genes is not a journal that specialises in pea genetics! The same applies to the abstract in which you should also remember that "the parts comprise the whole".
The punctuation hinders my understanding - for example where you use semicolons.
Remember that the word "data" is plural - ie "data are ...", not "data is ...".
Author Response
We thank the authou for these comments.
The sentence containing the word 'comprise' has been gorrected, and all agreements with 'data' are for a plural.
In addition to these changes additional grammatical errors have been corrected.